# Microwave-Assisted Synthesis of Azo Disperse Dyes for Dyeing Polyester Fabrics: Our Contributions over the Past Decade

**DOI:** 10.3390/polym14091703

**Published:** 2022-04-21

**Authors:** Alya M. Al-Etaibi, Morsy Ahmed El-Apasery

**Affiliations:** 1Natural Science Department, College of Health Science, Public Authority for Applied Education and Training, Fayha 72853, Kuwait; 2Dyeing, Printing and Textile Auxiliaries Department, Textile Research and Technology Institute, National Research Centre, 33 El Buhouth St., Dokki, Cairo 12622, Egypt; elapaserym@yahoo.com

**Keywords:** polyester fabrics, disperse dyes, ultraviolet protection factor, microwave irradiation

## Abstract

Organic reactions utilizing the microwave strategy have become able to conduct in shorter times, with higher yields, and are compatible with green chemistry protocols. In recent years, microwave technologies as an effective agent in organic synthesis have been successful utilized in textile industries and for the synthesis of dyes, especially disperse dyes. Herein, we present our contributions over the past decade through the use of microwave technology not only in the synthesis of new biologically active organic compounds and disperse dyes, but also the use of this effective, environmentally friendly technology in dyeing polyester fabrics as an alternative to conventional heating methods. We also demonstrate both the fastness properties and biological activities of the newly prepared compounds. In addition, we present the treatment of dyeing baths by reusing them again in the dyeing process, using microwave energy to achieve this goal, and this has environmentally friendly dimensions. Some of the possible utilizations of microwave irradiation have been presented in many different fields of chemistry. We recommend relying on this effective and environmentally safe technology instead of relying on conventional methods that take a lot of time, give low yields, and may have a negative impact on the environment.

## 1. Introduction

Azo dyes have been the most widely used synthetic dyes in the three last decades [1] because they are simple to make and have a wide range of industrial applications, including cosmetics, textile dyeing, and biological activities. Azo disperse heterocyclic dyes are commonly used for dyeing and printing polyester. Hence, they produce good brilliance colors, good tinctorial strength, and excellent fastness properties [2,3]. Polyester (PET) is one of the polymers containing ester bonds formed from the poly condensation reaction of dicarboxylic acid and diol. These fibers are less likely to wrinkle and have excellent washability and abrasion resistance. PET is the most hydrophobic fiber among ordinary fibers, and due to its compact form, the aqueous dyeing requires high energy to adsorb the disperse dyes. These fibers represent cheap and readily available raw materials with desirable properties, such as high strength, light weight, and excellent dyeability. Polyester fiber has a crystalline and compact structure and is highly hydrophobic. As a result, its aqueous dyeing is conducted at high temperatures and pressures, with disperse dyes. Polyester dyeing involves dissolving and re-dissolving disperse dyes, transferring soluble dyes from bulk solutions to the fiber surface, followed by the diffusion and adsorption of dye at the fiber surface, and diffusion from the surface into the interior of the fiber (*C.f.* Figure 1).

It is known that the availability of safe methods for preparing chemical compounds and the reduction of the use or even non-use of organic solvents harmful to the environment represent some of the most important requirements of green chemistry [4,5,6,7,8,9]. Azo dyes are widely used in the dyeing of textiles, leather, cosmetics, plastics, and food. In view of the toxicity, these dyes appear as a carcinogen towards the environment in the form of liquid waste. In textile wet processing techniques, the dye bath needs effective fundamental modifications that not only reduce effluent loading, but make textile processing environmentally friendly. Among these modifications is the use of modern techniques that make the dyeing process cost, time, and labor efficient as well as sustainable. Sustainable energy in the form of gamma radiation, ultrasound, infrared radiation, radio waves, ultraviolet radiation, and microwaves is used to improve the uptake ability of fabric, increase the color yield, and minimize the textile processing effluent load [10,11,12,13]. Fazal et al. [14] improved the dyeing behavior of polyester fabric via surface modification with ultrasonic treatment utilizing disperse Red 343, keeping in mind the regular use and good benefits of ultrasonic treatment. Fazal et al. [14] came to the conclusion that ultrasonic energy can be used to improve the dyeing behavior of other fabrics utilizing various dye classes [14]. Adeel et al. [15] reported that ultraviolet radiation can be utilized to enhance the color fastness properties and color strength without harming the chemical characteristics of polyester fabric via the application of disperse dye yellow 211. Ghaffar et al. reported that microwave radiation progresses the color asset of the dye solution of reactive blue 21 and increases the color fastness properties on cotton fabrics [16].

Microwave irradiation is commonly used to speed up a wide range of chemical reactions. In many circumstances, minutes of microwave irradiation are enough to accomplish reactions that would normally take hours. Thermal effects represent another well-known mechanism which allows microwave irradiation to accelerate chemical reactions. Increasing the frequency of molecular vibrations during microwave irradiation seems to accelerate these reactions. Rana et al. [17] reported that microwave heating enhances the rate of chemical reactions. This is due to its ability to considerably augment the temperature of a reaction. De la Hoz et al. [18] studied and presented some features that can be used to predict the possibility of optimizing reactions under microwave radiation by simple calculations of activation energy, enthalpy, and polarity. An environmentally friendly feature of using microwave energy is rapid heating to high temperatures in airtight containers, allowing for greater ease of reactions, as well as reduced or no solvent use, and this is important for the field of green chemistry [19,20,21,22,23,24,25,26,27,28,29,30,31,32,33,34]. This leads to the selectivity of the reaction. In this review article, we present our contributions over the past decade in using microwave technology to synthesize many new organic compounds and disperse dyes that possess biological, anticancer, and antioxidant activities in a very short time of a few minutes and with greater yields than the conventional preparation method. Moreover, the biological activities of the prepared compounds and dyes are presented. In addition to the use of microwave technology in dyeing polyester fabrics, which provides a much higher intensity of colors than conventional dyeing methods, this can reflect positively on the environment, as pollution rates are decreased. It is of value to mention here that the laboratory microwave oven used is a single mode cavity Explorer Microwave with a maximum power of 300 watts (CEM Corporation, Matthews, NC, USA) and irradiation was conducted in a heavy-walled Pyrex tube (capacity 10 mL for synthesis and 80 mL for dyeing processes (*C.f.* Figure 2).

## 2. Synthesis and Characteristics

### 2.1. Chemistry

It can be said that microwave technology offers a lot of applications and distinctive benefits, and we can also point out here that if we have a better understanding of the physical basis of the coupling mechanisms between microwave irradiations and matter, it is possible to extend the use of microwave technology to innovative scientific utilizations. We started as a research group in 2011 with the goal of using microwave heating in the synthesis of new disperse dyes through the reaction of hydrazine hydrate **5** with hydrazonocyanoacetate **4** [21]. Hydrazone **4** is formed under mild conditions by adding ethyl cyanoacetate **1** to diazoniun salt **2**. Furthermore, the ethyl group is represented by only two sets of two sp^3^ carbon signals in the ^13^C NMR spectrum of hydrazone **3**. Moreover, the results show that the hydrazone product is a 1:2 equilibrium mixture of *syn*-form of compound **3** and *anti*- form of compound **4** (Figure 1, Figure 3) [21,23].

Consequently, hydrazone **4** reacts with hydrazine hydrate **5** in ethanol to produce dye **6**. Using microwave irradiation, we discovered that compound **6** rapidly condenses with acetylacetone **7** to give the disperse dye **9** (*c.f*. Figure 2, Figure 4). Vishwakarma et al. explored the reaction of aminopyrazole with enaminones in the presence of 2 equivalent amounts of KHSO_4_ to afford the diarylpyrazolo [1,5-a] pyrimidines derivatives in aqueous medium water-ethanol (1:1) as a green solvent and under thermal conditions (60 °C for 1–4 h) in 71–79% yield [2]. Using microwave irradiation, compound **6** interacts with enaminones **10a**–**d** to produce the disperse dyes **13a**–**d** (*c.f*. Figure 3, Figure 5, Table 1).

The coupling of malononitrile **14** with diazonium salt **2** to yield compound **15** is one of the sequences employed in the synthesis of the disperse dye **20a**–**h** (Figure 4). Nuclear Overhauser effect (NOE) tests reveal that irradiation of the hydroxyl signal causes an enhancement of the aryl proton signal [22].

Hydrazine hydrate **5** refluxes with compound **15** to afford the disperse dye **16** [22]. Hydrazone **18** was obtained in high yield and reached 85% when coupling compound **17** with diazonium salt **2** in the presence of ethyl alcohol/sodium acetate. NOE experiments were carried out to aid in the formation of the structure of **18**. NOE experiments show that irradiating the NH signal at 12.1 ppm causes an enhancement of the aryl proton resonances at 7.39 and 6.80 ppm (Figure 4) [22]. The disperse dye **19** is produced by reacting hydrazone **18** with hydrazine hydrate through refluxing for four hours in the presence of ethanol. Irradiation of the NH signal at 11.94 ppm corresponding to compound **19** enhances the methyl proton signal at 2.36 ppm according to NOE difference measurements [22]. Dyes **16** or **19** readily condense with the enaminones **10a**–**d** through refluxing for one hour in the presence of acetic acid and sodium acetate to produce disperse dyes **20a**–**d** or **20f**–**h** (Figure 4) [22].

Using microwave irradiation, we were able to describe in 2012 the one-pot synthesis of compound **9** with higher outcome via hydrazone **4**, hydrazine hydrate **5**, and acetylacetone **7**. Disperse dye **9** possesses an ability to exist in isomeric structure **8**. Based on X-ray crystallographic structure determination, the occurrence of the isomeric structure **8** is excluded (*c.f.* Figure 5) [23].

Using microwave irradiation, we previously showed in 2011 [21] that compound **6** interacts with enaminones **10a**–**d** to produce the disperse dyes **13a**–**d**. We could also report in 2012 [23] that **13a**–**d** could be synthesized directly by using microwave heating for 5 min at 130 °C via the one-pot interaction of compounds **10a**–**d**, hydrazone **4**, and hydrazine hydrate **5** (*c.f*. Figure 6, Table 2) [23].

Although it is thought that these disperse dyes exist mostly in keto-structures **12a**–**d**, the predominance of compounds **13****a**–**d** might be attributed to stabilization of the products by hydrogen bonding between hydroxyl group OH and azo group -N=N- (*c.f.*
Figure 5) [22]. Altalbawy et al. reported that the reaction of aminoarylazopyrazole with a molar equivalent of arylidinemalonitrile in refluxing pyridine for four hours afforded the corresponding cyclized products pyrazolo [1,5-a] pyrimidine derivatives in 60–75% yields [8]. El-Apasery et al. explored that the reaction of arylhydrazonopyrazolones with enaminones in refluxing acetic acid for three hours afforded the 7-Phenyl-3-arylazo-pyrazolo[1,5-a]pyrimidin-2-one derivatives in 51–81% yields [13].

It is well known that dye **25** was synthesized from diketones **21** with various diazonium chloride **22** to produce substituted arylazodiketones **23**, which can then be condensed with cyanoacetamide derivatives **24** using either a conventional or microwave heating approach [35,36,37,38,39,40,41,42] (Figure 7).

The interaction between diazonium chloride and pyridones, on the other hand, is used in the second reaction route for the synthesis of these azo dyes. In 2014 [24], by using microwave irradiation at 160 °C for 20 min, we reported a three-component condensation of methyl propionylacetate **26** as *β*-ketoesters, ethyl cyanoacetates **1**, and ethyl amines **27** to yield pyridine **28**. It is worth mentioning that in 2013 [20], we prepared compound **28** using conventional heating for 6 h. This compound possesses two tautomeric structures, which quickly equilibrate in solution (Figure 8).

As illustrated in Figure 8, pyridine **28** could be coupled with various diazonium salts **22** to produce the disperse dyes **30a**–**h** that exist in the hydrazone tautomeric form based on X-ray crystallographic structure determination (Figure 6, Figure 7 and Figure 8).

### 2.2. X-ray Crystallographic Structure Determination

X-ray crystallography is a tool for determining the atomic and molecular structures of crystals. This method revealed the structure and function of many molecules, such as dyes, nucleic acids, proteins, drugs, proteins, and vitamins. The underlying principle is that crystal atoms diffract an X-ray beam in many specific directions. By measuring the angle and intensity of these diffracted beams, it could create a three-dimensional image of the electron density in the crystal. From this electron density image, we can determine the average position of atoms in the crystal, their chemical bonds, their bond length, and their bond angle. Crystallographic data for the structures of compounds **4**, **9**, **13c**, **30a**, **30b**, and **30g** reported in this paper have been deposited with the Cambridge Crystallographic Data Centre (CCDC) as supplementary publications Nos. 848619, 848620, 871092, 925725, 925785, and 930799. The green, red, and blue spheres represent atoms of carbon, oxygen, and nitrogen, respectively. The white spheres represent hydrogen, which were determined mathematically rather than by the X-ray analysis.

X-ray crystallographic structure determination has solved for us three important problems in confirming the chemical composition of some chemical compounds. Figure 3 confirms that the NH proton would be deshielded in the ^1^H NMR spectrum due to possible hydrogen bonding with the carbonyl ester group, and the main isomer is anticipated to have the *anti*-form of compound **4** [23].

Figure 4 and Figure 5 confirm the chemical structure of compounds **9** and **13**, thus excluding the occurrence of the chemical structure of compounds **8** and **1****1** [23]. Figure 6, Figure 7, Figure 8 and Figure 9 confirms the chemical composition of the dyes **30a**, **30b**, and **30g** and their presence in the hydrazone-form with the exclusion of their presence in the azo-form [24].

## 3. Dyeing

Using microwave heating at 130 °C, the disperse dyes **6**, **9**, **13a**–**d**, **16**, **19**, and **20a**–**h** were utilized for dyeing polyester fabrics with shades 1–6%, yielding a variety of color shades. The results in Table 3 and Table 4 indicate the microwave irradiation’s efficiency, which resulted in a considerable increase in dye uptake and dyeing rate. The amount of dye lost each time determines whether dyebaths may be reused.

We planned to reuse the dye bath in this investigation by increasing the dyeing period without adding any new dye. Both of the color strengths of dyeing baths for 60 min and the reused dyeing baths for 90 min are shown in Table 3. The color consistency of dyeing remained satisfactory after reuse of the dye bath, which is difficult to identify with the human eye.

K/S values expressed the color strength, which was measured at the maximum wavelength λ_max_. K/S was performed by utilizing the Kubelka–Munk equation [1,21].
(1)$$K/S\;=\;\frac{{\left(1\;-\;R\right)}^{2}}{2R}\;-\;\frac{{\left(1\;-\;R_{0}\right)}^{2}}{2R_{0}}$$
where R is the decimal fraction of the reflectance of the dyed fabric; R_0_ is the decimal fraction of the reflectance of the not dyed fabric; K is the absorption coefficient; and S is the scattering coefficient.

It is also clear from the results in Table 3 that we have obtained values for color strength K/S at first dyeing process. We also obtained values for color strength K/S re-using the dyeing baths process, and when comparing K/S values of the two processes we found that the differences in K/S values that we obtained from reusing the dyeing baths a range from 0.24% to 46% of the K/S values for the first dyeing process. We can report that we obtained K/S with percentages ranging from 99.76% to 54% reusing the dyeing baths when compared to the K/S of first dyeing. Table 4 clearly shows that the color strength obtained with dye **20d** is substantially greater than that obtained with dyes **20a**–**c** and **20h**.

Compounds **30a**–**h** were used to dye polyester fabrics using a high-pressure, high-temperature dyeing process at 130 °C with shade 2%.

The color of the dyed fabrics ranged in color from yellow to dark orange. After that, the dyeing characteristics of polyester fabrics were assessed in terms of their fastness properties. The **30a**–**h** hues had a strong affinity for polyester materials, according to the K/S estimates in Table 4, and all color strengths were usually favorable.

The CIELAB (Color space defined by the International Commission on Illumination (CIE) in 1976) psychometric coordinates *L**, *a**, and *b** represent the color hues and were estimated for the color of the dyed example, where *L** represents lightness, *a** represents the red–green axis; and *b** represents the yellow–blue axis [20,21,22,23,24,25,26,27,28,43,44].

The total color difference Δ*E** was measured by using an UltraScan Pro (Hunter Lab, Reston, VA, USA) 10° observer with D65 illuminant, d/2 viewing geometry, and measurement area of 2 mm. The total color difference ∆*E** between the sample and the standard was calculated using the following equation:(2)$$\Delta E^{*}\;=\;\sqrt{{\left(\Delta L^{*}\right)}^{2}\;+\;{\left(\Delta a^{*}\right)}^{2}\;+\;{\left(\Delta b^{*}\right)}^{2}}$$
where Δ*L**, Δ*a**, and Δ*b** are the derivatives of corresponding parameters, respectively.

Incorporating electron-donating groups lowered brightness while introducing electron withdrawing groups into the benzene ring increased lightness and brightness, resulting in the colors **30e**–**g** being lighter and brighter than the **30b**–**d**.

In order to confirm the efficiency of the microwave dyeing process, El-Apasery et al. [3] presented a study in which they compared the results of dyeing of polyester fabrics with C.I. disperse red 60 using microwave technology with the results of conventional dyeing in the presence of a carrier. Data listed in Table 5 show an increase in K/S values for microwave and ultrasonic dyeing compared to conventional dyeing.

This might be due to microwave and ultrasound waves facilitating a dye–fiber contact and accelerating the rate of diffusion of the dye inside the fiber by breaking the boundary layers covering the fiber and raising the interaction between dye molecules and fibers through the cavitation phenomenon [3].

The color coordinates listed in Table 5 show that the dye has a good affinity for polyester fabrics at the indicated temperatures, giving it a bright and intense red tint. The color of a sample dyed by microwave or sonication is darker than the same sample dyed with the same dye using conventional dyeing methods.

## 4. Fastness Properties

The fastness properties of the dyed samples, such as perspiration, washing, and light, were tested according to the tests of the American Association of Textile Chemists and Colorist [21]. The data listed in Table 6 show that the fastness data obtained by measuring the color fastness properties of polyester fabrics dyed with dyes **6**, **9**, and **13a**–**d**. Table 6 shows the results of color fastness to washing, light, and perspiration, where the ratings for color fastness to light and washing were good and very good while the rating for color fastness to perspiration was excellent [23].

The data listed in Table 6 show the fastness data obtained by measuring the color fastness properties of polyester fabrics dyed with dyes **16**, **19**, and **20a**–**h**. The fastness ratings recorded in Table 6 show excellent perspiration fastness as well as excellent washing fastness with respect to all of the tested dyes except dye **20d**. The light fastness of polyester dyed fabrics displayed moderate fastness. The light fastness is significantly affected by the nature of the substituents in the diazonium component. Hence, electron donating groups on this moiety should increase the fading rate while electron withdrawing groups should decrease the rate. This proposal is in agreement with the observed results (Table 6) which demonstrate that the presence of a methyl group in dyes **20b** and **20g** causes a decrease of light fastness to 3. On the other hand, the chlorine atom in dyes **20c** and **20h** is associated with an increase of light fastness to 4 and 6, respectively [22].

The data listed in Table 6 show the fastness data obtained by measuring the color fastness properties of polyester fabrics dyed with dyes **30a**–**h**. The fastness ratings are recorded in Table 6 show very good fastness levels to perspiration and excellent fastness levels to washing. The light fastness of polyester dyed fabrics displays moderate fastness. The light fastness is significantly affected by the nature of the substituents in the diazonium component. The inclusion of electron-withdrawing (bromine or chorine or nitro) substituents improves the light fastness to (3–4), (3–4), and (5) in dye **30e**, **30f**, and **30g**, respectively [24].

Generally, the results obtained showed that the dyed fabric could have good fastness in terms of the following points: (i) Good diffusion of dye molecules in the fibers inside the fabrics. (ii) The size of the dye molecule is considered to be relatively large. (iii) There are no solubilizing groups that affect the solubility and detergency of fabric dyeing [24].

## 5. Antimicrobial Activities

The antimicrobial activities of the synthesized dyes **4**, **6**, **9**, and **13a**–**d** were tested using the agar well diffusion method against a variety of bacteria and fungi. The listed data in Table 7 show promising positive antimicrobial activities. Compounds **4** and **6** had high antibacterial activity against Gram-positive bacteria, whereas the others had moderate to poor antibacterial activities. After six days of incubation, all the tested compounds reduced the development of *Candidia albicans*.

Table 8 demonstrates that all of the tested dyes had strong positive antibacterial properties against four of the investigated pathogens based on inhibition zone diameter data for the disperse dyes **30a**–**g**. Even after 120 h of incubation, disperse dye **30a** showed cytolytic impact, with no growth reported in the inhibition zone for all tested microorganisms.

We were able in 2020 to show that treated polyester dyed fabrics of dye **30h** with TiO_2_ nano particle size exhibited strong antifungal activity against *Aspergillus flavus* and *Penicillium chrysogenum*, with inhibition zones of 21 and 19 mm, respectively [27]. This is consistent with recent findings [28,29,30] that suggest it is possible that the treatment of fabrics with oxides in nano particle size will provide several properties, including the anti-bacterial property of those treated fabrics.

## 6. Antioxidant Activities (DPPH Radical Scavenging Activity)

The methanol solution of the 2,2-diphenyl-1-picrylhydrazyl (DPPH) radical was prepared and stored at 10 °C. A methanol solution of the test disperse dyes were prepared. A 40 µL aliquot of the methanol solution was added to 3 mL of DPPH solution. The decrease in absorbance at 515 nm was determined continuously, with data recorded at 1-min intervals until the absorbance stabilized (16 min). The absorbance of the DPPH radical in the absence of the antioxidant (control) and the reference compound ascorbic acid was also assessed. The percentage inhibition (PI) of the DPPH radical was calculated according to the formula:
PI = [{(*A*C − *A*T)/*A*C} × 100](3)
where *A*C represents the absorbance of the control at t = 0 min and *A*T represents the absorbance of the sample + DPPH att = 16 min [26].

We tested the disperse dyes’ antioxidant properties in vitro for dye **30h** using the DPPH free radical scavenging activity. Table 9 and Figure 9 show that dye **30h** has a higher antioxidant activity than ascorbic acid, which has an IC_50_ of 14.2, with an IC_50_ of 64.5 (*c.f.* Figure 10) [26].

Figure 9 reveals that the antioxidant capacity of dyes **30h** was good with an IC_50_ value of 65.4 when compared with ascorbic acid (IC_50_ value 14.2), indicating that the related molecule is a promising antioxidant.

## 7. In Vitro Cytotoxicity Activities

In addition, we reported in 2019 that we were investigating the anticancer efficacy of the dye **30h** against MCF-7 cells (breast cancer), HepG-2 cells (hepatocellular carcinoma), A-549 cells (lung carcinoma), and HCT-116 cells (lung carcinoma) (colon carcinoma), using Cisplatin and Imatinib as the reference drugs (*c.f.* Figure 11, Figure 12, Figure 13 and Figure 14).

The IC_50_ was determined using different doses of disperse dye **30h**. Table 9 and Figure 11, Figure 12, Figure 13 and Figure 14 demonstrate that dye **30h** had high activity, with values of 62.2(±4.1) (MCF-7), 23.4 (±1.2) (HepG-2), 53.6 (±5.8) (A-549), and 28 (±1.9) (HCT-116), respectively [26].

## 8. Hazards Risk of Azo Dyes

It is well known that azo dyes facilitated the development of the chemical industry and it is well known that the hazard risk of azo dyes is not inherent, but rather due to their by-products generated after oxidative reductions or enzymatic cleavage. For example, substituted anilines are pollutants and harmful to our environment, cause cancer, allergies, and many human diseases. It is necessary to develop effective strategies through which to control the spread of contamination of toxic azo dyes and aromatic amines, so that we can reduce cancers and multiple related diseases. We would like to stress here that the discharge of liquid waste from the textile and dyeing industries into water bodies causes major environmental health concerns. Decolorization has become of great global scientific importance. During the past decade, many physical and chemical decolorization techniques have been reported in the textile industries, but many of them have not been implemented due to their high cost, low efficiency, and inapplicability to a wide range of dyes. Therefore, the ability of microorganisms to perform the decolorization of a wide range of dyes has received wide attention as it represents a cost-effective method for removing these pollutants from the environment. It is worth mentioning here that since 2011, we have worked to develop an effective decolorization strategy for some azo dyes and implemented it through the use of economical and environmentally safe methods [4,21].

## 9. Gap Analysis of Microwave Technology and Recommendations

Microwave heating is a highly effective heating source in chemical reactions, as it can speed up and increase the reaction rate. Moreover, it provides better productivity and uniform heating, while the use of solvents in a chemical reaction can be reduced or eliminated, while providing greater reproducibility of chemical reactions compared to conventional heating [43,44,45,46,47,48,49,50,51,52,53,54,55,56,57,58,59,60,61,62,63,64,65,66,67,68,69,70,71,72,73,74,75,76]. However, this technique faces some limitations, including the fact that a sudden increase in temperature may result in undesirable products, it is difficult to conduct temperature sensitive reactions, and it presents limited industrial applications. Thus, care must always be taken during the process. Despite these limitations, we can claim with certainty that microwave chemistry has opened many new avenues for organic synthesis. In addition, many reactions that were once nearly impossible or that resulted in low throughput and time-consuming reactions can now be performed quickly, efficiently, and safely in a matter of minutes. Microwave chemistry has changed the world of organic chemistry and drug discovery. Therefore, it is recommended to adopt this efficient and environmentally safe technology instead of relying on conventional methods that take hours to days and often have a negative impact on the environment. We can expect that in the next few years the use of microwave heating technology will increase in industry and become a very popular and inexpensive technology.

## 10. Conclusions

In this review article, we highlight that the microwave-assisted synthesis of new disperse dyes is feasible by reducing the reaction time from hours to minutes, sometimes up to seconds, as well as using lower proportions of organic solvents. The possibility of using microwave energy to dye polyester fabrics with excellent efficiency was demonstrated. The biological activity of new disperse dyes has been reviewed and this gives these dyes an added value. Some limitations are presented by this technique, although its many benefits compared to traditional heating methods were also clarified. We can point out that microwave chemistry has opened up many new routes for organic synthesis. We can perform many reactions that were almost impossible, or that formerly resulted in low productivity, quickly, efficiently, and safely. We therefore recommend this effective technology and anticipate that in the coming years microwave heating technology will be widely used in industry and thus become an inexpensive and widespread technology.

## Data Availability

Not applicable.

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
