# Peer review of "Microwave-Assisted Synthesis of Azo Disperse Dyes for Dyeing Polyester Fabrics: Our Contributions over the Past Decade"

_polymers, 2022, doi:10.3390/polym14091703_

Round 1

Reviewer 1 Report

The work reported by authors is good however following changes needed to improve the quality of the work.

1- in introduction give one paragraph about using modren tools in field of synthetic dyes such as radiation treatment. authors can take help from work of fazal et al 2021  global nest journal,  fazal et al 2020  polish j env studies, ghaffar et al 2020 polish j env studies,  samreen gul khan et al j mexican chem society,  shumaila kiran et al  j natural fibers,  global nest journal, Adeel and bhatti et al saudi j chem soc,  indian j  of textile and fibre research,  majid muneer et a oxidation communicvations 2014 2015 etc.

2- in discussion add FTIR  of undyed and  dyed work if possible

3- show SEM  analysis if possible before and after coloration

4-  give some reference with reasoning for justification of your discussion.

5- do not start any seentence with digit and follow SI units to show   unit of work

6- see some formating style of references as per  journal

7-   give comparison and future prospects in 2-3 lines in conclusion as well as in abstract

Author Response

Please find our attached response file

Reviewer 2 Report

The paper reviews the authors' results for the last years. It summarizes their achievement in microwave assist disperse dyes synthesis and in the dyeing of polyester textile with them. The results are compared with the ones obtained with conventional heating methods. The dyes' biological activity was also presented.

The manuscript will be of interest to readers of Polymers, but some minor revisions should be made before publishing.

In section Dyeing:

Line 204: As the dyeing with the compounds (30a-h) is different from the dyeing with other dyes, separate the paragraph into a new line in the text. 

In addition, the authors have to specify the shade used for dyeing with dyes 30a-h.

The explanation in the paragraph (Line 199 ÷ 201) and the title of the third column in Table 3 are unclear.

The text mentions the differences in colour strength K/S in %, but the unit of colour strength K/S is not %. At the same time, in Table 3, the title of the third column is - Differences Shades %. Please specify what these values mean 0.24% to 46% and 99.76% to 54%.

Author Response

Please find our attached response file.

Reviewer 3 Report

The authors present an original study about the application of microwave technology on azo disperse dyes synthesis and their use in PET dyeing. The manuscript is clear, interesting and very rich on the fundamental level. However, it lacks a lot of details and data, especially about the dyeing section. So, several comments should be addressed in particular the following points:

- English mistakes should be revised carefully.

- The polymer science aspect is not developed in this paper. The authors should revise their paper to meet the journal requirements.

- Page 1/Line 40: “Brittany [16] reported… ” This sentence should be revised.

- Page 9/Dyeing

The characteristics of textile material should be given.

The reagents used in the dyeing process and textile preparation should be given.

The authors used an innovative process for dyes synthesis and PET dyeing. So, they should add a schema of the used device with details about its components.

How is the agitation performed in the dyeing bath?

What is the liquor ratio used in the dyeing process and how is it kept constant?

How did the authors determine the dyeing temperature?

What is the dyeing duration?

All details about the dyeing process should be included in the paper.

The reduction cleaning process of dyed PET fabrics should be added.

- In order to confirm the efficiency of this innovative dyeing process, the dyeing results using microwave technology should be compared to those of conventional dyeing at 130°C and to those of conventional dyeing at 100°C in the presence of carrier.

- Page 10/Line 298: The substantivity and the interactions between azo disperse dyes and PET should be discussed with details.

- Page 10/Line 298: The method used to determine K/S and CIELab parameters should be given with details.

- Page 10/Table 3: The difference between shades should be appreciated in terms of ∆E or ∆ECMC parameters.

- Page 1/Line 216: Standardized procedure for determining colour fastnesses as well as corresponding results should be included.

- Page 11/Table 5: It can be seen that dyes 4 and 6 have a serious antibacterial activity. So, if these azo dyes will be applied in PET dyeing, the generated effluents cannot be treated with biological process. What is the solution in this situation?

- The cytotoxicity study was only applied to dye 30h and what about the others synthesized azo dyes?

- It is well known that hazard risk of azo dyes is not themselves, but specially their by-products generated after oxidative or enzymatic cleavage. So, it is important to study the toxicity of these byproducts of synthesized azo disperse dyes like substituted anilines.

 - Conclusion requires more development.

- Many references should be revised.

Author Response

Please find the attached response file

Reviewer 4 Report

(1) Routine review paper

(2) Technical content was extracted from different published papers

(3) No critical comment from the authors was given

Author Response

Please find attached our response file

Round 2

Reviewer 1 Report

ACCEPTED FOR PUBLICATION

Author Response

Thank you 

Reviewer 2 Report

The authors have responded to my comments.

Author Response

Thank you

Reviewer 3 Report

The following remarks should be taken into account :

  • For the added sections, some English mistakes should be also revised.
  • The authors should show the position of the revised parts in the text.
  • The method used to determine K/S. The authors measured K/S at λmax, or they used another method ? please, explain that.
  • The conditions to determine CIELAB parameters as well as the used devise should be added.
  • Page 11/Line 275: in the equation L*, a*, b*, ∆E* should be revised. The same remark is for Table 5.
  • It is so important that the results of colorfastnesses should be added and discussed. If not the paper will be rejected.

Author Response

Please find attached our response file

Reviewer 4 Report

No further comment

Author Response

Thank you

Round 3

Reviewer 3 Report

After revision, I think that the paper is now suitable for publication. An advice for the authors. For their future papers, they must show with details the position of the revised parts in the text. I mean the page and the line.